# OpenReview forum: "How Many Van Goghs Does It Take to Van Gogh? Finding the Imitation Threshold"
_ICLR.cc/2025/Conference — Submitted to ICLR 2025_

### Official Review · Reviewer_hEno · 2024-10-28

**Soundness:** 2
**Presentation:** 3
**Contribution:** 2
**Rating:** 5
**Confidence:** 5

**Summary:**

The author experimentally  explored the number of data samples required for a text-to-image model to generate outputs that exceed the imitation threshold for a given concept, meaning that truly convincing imitation can be considered to exist. The author proposed MIMETIC2 to address the high computational cost of existing approaches.

**Strengths:**

The authors try to determine the number of training samples required for a text-to-image model to truly imitate a specific concept, and the problem is well-motivated.

The manuscript has clear logic, from the definition of the problem, rigorous assumptions statement, and experimental verification.

The authors discuss the limitations of the current work in assumptions and problem simplification and clarify valuable future work

**Weaknesses:**

1. The findings are based on limited experiments, which constraints the insights of the results. Under the strict assumptions set by the author, giving quantitative relationship between Concept Similarity and Concept Frequency or an imitation threshold that generalizes to other datasets, such as Laion-400m [1], will be more meaningful. For example, if the quantitative relationship shows exponential decay or exponential increase, it would reflect the possibility of whether a small sample is sufficient to raise infringement issues in a social sense, giving the model trainer a guidance on data usage.

2. The experimental results show limited insights, the types of datasets and models are limited, and are tailored to meet the assumptions pointed out by the author. Although the author lists the challenges encountered in the experiment, such as outliers, there lacks a discussion about  feasible solutions. In general, the scenarios considered are too few and too strict, which is far from the practicality of achieving the goal of the paper demonstrated in Section 1, such as judging whether the infringement claim is established and guiding developers to avoid infringement.

3. It is recommended that the author add optimal approach to find the imitation threshold (Pearl (2009)) as a comparison to verify the correctness of the imitation threshold calculated by the proposed MIMETIC2 method.

[1] Schuhmann, C., R. Kaczmarczyk, A. Komatsuzaki, A. Katta, R. Vencu, R. Beaumont, J. Jitsev, T. Coombes, and C. Mullis (2021). “LAION-400M: Open Dataset of CLIPFiltered 400 Million Image-Text Pairs”. In: NeurIPS Workshop Datacentric AI. FZJ-2022-00923. Jülich Supercomputing Center

**Questions:**

1. I hope the author can further clarify the given experiment found that the imitation threshold is 200-600 images. Is it a generalized result in a wide range of scenarios, such as other datasets Laion-400m [1], so as to truly provide valuable guidance for judging whether the infringement claim is established and guiding developers to avoid infringement?

2. If the imitation threshold is not reached, can it be considered that there is no infringement?

3. Can the quantitative relationship between Concept Similarity and Concept Frequency be given?

4. Add optimal approach to find the imitation threshold (Pearl (2009)) as a baseline comparison to verify the correctness of the MIMETIC2 method will be better to evaluate whether the improvement in time efficiency lead to a loss in accuracy of imitation threshold  estimation.

[1] Schuhmann, C., R. Kaczmarczyk, A. Komatsuzaki, A. Katta, R. Vencu, R. Beaumont, J. Jitsev, T. Coombes, and C. Mullis (2021). “LAION-400M: Open Dataset of CLIPFiltered 400 Million Image-Text Pairs”. In: NeurIPS Workshop Datacentric AI. FZJ-2022-00923. Jülich Supercomputing Center.

---

> ### Author Response · Authors · 2024-11-20
> **Author Response**
>
> We thank the reviewer for their time and effort to review our work. We are glad that the reviewer found our work to have a clear logic, assumptions clearly stated, and have a rigorous experimental verification. We address the limitations as follows:
>
> > 1. Is it a generalized result in a wide range of scenarios, such as other datasets Laion-400m [1], so as to truly provide valuable guidance for judging whether the infringement claim is established and guiding developers to avoid infringement?
>
> We agree with the reviewer that our claims about the imitation thresholds of around 200-600 images should generalize to other datasets in order for them to be valuable to the community. While our main contribution is a generalizable methodology that will be applicable to all model/data pairs in the future, we rigorously test whether our thresholds are generalizable. For this, we experimented with all the models whose training data was open-source (SD1.1, SD1.2, SD1.3, SD1.4, SD1.5, and SD2.1), with four different domains and found the threshold to be within this range (Table 3 in Section 6 and Table 6 in Appendix G). To answer the reviewer’s question, we also ran experiments with LAION-400M dataset on which a latent diffusion model was trained.
>
> |Dataset| Model| Celebrities|Politicians|Classical|Modern|
> |-|-|-|-|-|-|
> |LAION-400M|LD|648|309|219|282|
>
> We report the imitation thresholds in the table above. We find that all the thresholds are around the 200-600 range we found for LAION-2B and LAION-5B, and we therefore hope that our threshold will generalize to other datasets as well. We have added this experiment to the paper in Appendix A (L. 864-880).
>
> > 2. If the imitation threshold is not reached, can it be considered that there is no infringement?
>
> Based on our results showing the relation between a concept’s frequency and its imitation score (Figure 5 in the main paper and Figures 14-28 in the Appendix), we conclude that if the frequency of a concept is below the imitation threshold, it is unlikely for a model to be able to imitate it. Therefore, if the imitation threshold is not reached, infringement is unlikely.
>
> > 3. Can the quantitative relationship between Concept Similarity and Concept Frequency be given?
>
> We believe that the reviewer is interested in understanding the general mathematical relationship between concept frequency and similarity (which we empirically plot in Figure 5 in the main paper and Figures 14-28 in the Appendix). Our work does not focus on determining the precise mathematical relationship, and instead focuses on identifying the threshold that leads to imitation. However, previous works have found a “log-linear relationship” between frequency and similarity: https://arxiv.org/abs/2404.04125, https://arxiv.org/abs/2211.08411.
>
> Kindly let us know if you meant something different by 'quantitative relationship,' and we would be happy to provide clarification.
>
> > 4. Add optimal approach to find the imitation threshold (Pearl (2009)) as a baseline comparison to verify the correctness of the MIMETIC2 method
>
> Even though we would like to verify the correctness of MIMETIC^2 using the optimal approach as a baseline, this is an infeasible experiment as we note in Section 3 of the paper. For the optimal experiment, we would need to train O(log m) models, where m is the total number of available images of a concept Z^j. So if a concept has 100,000 images, then we need to train log_2(100, 000) = 16 models to optimally estimate the imitation threshold, which would cost about $10 million. This is way beyond the computational capability of our academic lab (and even industry in most cases).
>
> Therefore, we verify the correctness of the imitation thresholds in a tractable manner using an approximate method. Concretely, we calibrate MIMETIC^2 using one of the sets in each domain (Celebrities for Human Faces and Classical art styles for Art Style) and test the calibrated approach on the other set (Politicians for Human Faces and Modern art styles for Art Style). In essence, the sets that we use to calibrate MIMETIC^2 on, serve as the “train datasets” and the other sets serve as the “test dataset” (these sets are mutually exclusive). Table 3 in Section 6 shows that the imitation thresholds MIMETIC^2 finds for “train” and “test” sets in each domain are very close, indicating that the imitation thresholds MIMETIC^2 finds are accurate.
>
> We note this on Lines 456-467 in Section 6 of the paper.
>
> However, to answer the reviewer’s question, we are conducting a finetuning experiment that is the closest to the optimal experiment that is tractable. We will update the rebuttal with the result in a few days.

---

> > ### Comment · Reviewer_hEno · 2024-11-27
> >
> > Thank you for your response. My primary concern is whether the experimental results presented in the paper hold practical insights. The authors claim that "if the frequency of a concept is below the imitation threshold, it is unlikely for a model to be able to imitate it. Therefore, if the imitation threshold is not reached, infringement is unlikely." However, judgments for copyright infringement—a severe and precise legal matter—on "unlikely" outcomes are problematic. Similarly, the main result of the paper, which identifies the imitation threshold of these models to be in the range of 200–600 images, still represents an uncertain range.
> >  I appreciate the authors' work on new experiments on the Laion-400M dataset. However,  the imitation threshold for celebrities is reported as 648, which falls outside the claimed 200–600 range. This further suggests that the main insights provided in the paper may be limited to current widely-used models or datasets, reducing their broader applicability for the future.
> >
> > Furthermore, there are some technical limitations to determining the imitation threshold. For example, the MIMETIC^2 rely on cosine similarity as the imitation score, which cannot fully capture the degree of imitation between images. Consider a scenario where image A is a scaled-down version of image B, with the remaining areas filled with some random color. In such cases, the cosine similarity between the two images might be low, but image A could still constitute an imitation of image B. This limitation casts doubt on the proposed method in correctly determining the imitation threshold.
> >
> > In summary,  the current results are not sufficient to support the authors' claim that it "provides an empirical basis for copyright violation claims and acts as a guiding principle for text-to-image model developers." Achieving this goal would require more rigorous approaches to determining imitation thresholds and more definitive insights.

---

> > > ### Author Response · Authors · 2024-12-03
> > > **Author Response**
> > >
> > > We thank the reviewer for their response to our rebuttal. We are glad that the reviewer found our experiment with LAION-400M dataset strengthening the paper. We address the limitations below:
> > >
> > > > However, judgments for copyright infringement—a severe and precise legal matter—on "unlikely" outcomes are problematic.
> > >
> > > Contrary to the case of being a precise legal matter, copyright infringements are usually subjective interpretations with courts assessing factors like originality and substantial similarity on a case-by-case basis. Notable examples include:
> > >
> > > 1. _Lenz v. Universal Music Corp. (2015)_: Stephanie Lenz posted a 29-second video of her child dancing to Prince's "Let's Go Crazy." Universal issued a takedown notice, leading Lenz to sue, arguing fair use. The court held that copyright holders must consider fair use in good faith before issuing takedown notices, highlighting the subjective nature of fair use assessments.
> > >
> > > 2. _Mannion v. Coors Brewing Co. (2005)_: Photographer Jonathan Mannion sued Coors for allegedly copying his photograph of basketball player Kevin Garnett. The court identified three aspects of originality in photography—rendition, timing, and creation of the subject—emphasizing the subjective evaluation of these elements in determining copyright protection.
> > >
> > > 3. _Cariou v. Prince (2013)_: Artist Richard Prince used Patrick Cariou's photographs in his artwork without permission. The court ruled that 25 of Prince's 30 works were transformative and thus fair use, underscoring the subjective analysis involved in determining whether a work is transformative.
> > > These cases illustrate that copyright infringement determinations often depend on subjective judicial interpretations of factors like fair use, originality, and substantial similarity. Therefore our imitation thresholds can help guide some of these interpretations, not make the final decision.
> > >
> > > > I appreciate the authors' work on new experiments on the Laion-400M dataset. However, the imitation threshold for celebrities is reported as 648, which falls outside the claimed 200–600 range. This further suggests that the main insights provided in the paper may be limited to current widely-used models or datasets.
> > >
> > > We would like to point out that the range we mentioned 200-600 are for different models and different domains. We do not claim that this range will hold for every model and dataset (we mention this at L 516-517 and 523-524 in the updated paper draft). Therefore, our experiments with the LDM model trained on LAION-400M does not violate any of the claims we made. Instead, it provides an additional data point, which strengthens our results and provides an additional signal our estimated imitation threshold generalizes between similar models.
> > >
> > >
> > > > For example, the MIMETIC^2 rely on cosine similarity as the imitation score, which cannot fully capture the degree of imitation between images. Consider a scenario where image A is a scaled-down version of image B, with the remaining areas filled with some random color. In such cases, the cosine similarity between the two images might be low, but image A could still constitute an imitation of image B.
> > >
> > > This particular example will be of no concern to our face embedding models, because they first extract the face of a person from an image before computing its face embedding. And therefore even if there are random colors on the edges, that will not impact the accuracy of the face embedding model and thereby not impact the imitation score.
> > > While our imitation scores rely on cosine similarity, it includes many different steps to make it robust to these kinds of problems. We point the reviewer to Section 5.2 in the paper where we detail our exact algorithm for computing the imitation score.

---

### Official Review · Reviewer_97Vw · 2024-11-04

**Soundness:** 3
**Presentation:** 3
**Contribution:** 2
**Rating:** 6
**Confidence:** 4

**Summary:**

The paper formalizes and investigates a novel problem of Finding the Imitation Threshold (FIT), which aims to determine the imitation threshold for two representative image types: human faces and art styles. Comprehensive empirical analysis of Stable Diffusion models and their respective pre-training datasets reveals that the imitation threshold for these models ranges between 200 and 600 images, depending on the setup. The estimated thresholds have important implications for users and developers, potentially serving as a basis for copyright and privacy complaints.

**Strengths:**

* The investigated problem of imitation in text-to-image models is both prevalent and significant.

* By introducing novel methods for estimating concept frequency, the proposed MIMETIC2 is carefully designed to minimize the influence of confounding factors.

* This paper marks the first effort to provide precise estimates of the imitation threshold, offering implications that could benefit both technical advancements and policy-making.

**Weaknesses:**

* While this paper represents a novel attempt to determine the imitation threshold of concepts in text-to-image models, its technical implications are unclear. The memorization phenomena in LLMs [1] and VLMs [2], along with their training dynamics, are well-explored areas. Although the paper introduces concept frequency and imitation score estimation methods to establish a more precise threshold, it is not evident what technical insights this threshold provides. Apart from conceptual policy-making related implications (i.e., informing text-to-image model developers what concepts are in risk of being imitated, and serving as a basis for copyright and privacy complaints), how can this metric possibly be adopted to advance imitation mitigation strategies, such as those outlined in Appendix A? This is an important and practical issue that must be clearly explained.

* Additionally, the analysis presents conflicting explanations for threshold differences. In Line 347, the authors attribute the higher threshold in SD2.1 to its larger LAION-5B pre-training dataset, while in Line 350, they suggest differences in text encoders between SD2.1 and SD1.5 as the key factor. This raises an important question: what is the primary driver of these threshold differences—dataset size or text encoder architecture? Moreover, the paper does not explore other potentially influential factors, such as model size, which limits the comprehensiveness of the analysis.

[1] Kushal Tirumala, Aram Markosyan, Luke Zettlemoyer, Armen Aghajanyan. "Memorization Without Overfitting: Analyzing the Training Dynamics of Large Language Models." NeurIPS 2022.

[2] Jie Ren, Yaxin Li, Shenglai Zeng, Han Xu, Lingjuan Lyu, Yue Xing, Jiliang Tang. “Unveiling and Mitigating Memorization in Text-to-Image Diffusion Models Through Cross Attention.” ECCV 2024.

**Questions:**

Please refer to the Weaknesses section and address the concerns listed.

---

> ### Author Response · Authors · 2024-11-20
> **Author Response**
>
> We thank the reviewer for their time and effort in reviewing our paper. We are glad that the reviewer agrees that our paper marks the first effort to provide an estimate of the imitation threshold, which is both a significant and underexplored problem. We address the limitations as follows:
>
> > 1. While this paper represents a novel attempt to determine the imitation threshold of concepts in text-to-image models, its technical implications are unclear. Apart from conceptual policy-making related implications (i.e., informing text-to-image model developers what concepts are in risk of being imitated, and serving as a basis for copyright and privacy complaints), how can this metric possibly be adopted to advance imitation mitigation strategies, such as those outlined in Appendix A?
>
> Our results on estimating the imitation threshold (Figure 5 in the main paper and Figures 14-28 in the Appendix), indicate that if the frequency of a concept is below the imitation threshold, it is unlikely for a model to imitate it. Therefore, the implications of our results are relevant to the data curation practices. of text-to-image models like Stable Diffusion or Dalle: We hypothesize that if developers restrict the frequency of concepts they want to avoid imitating by keeping their frequency below the threshold, trained models are unlikely to imitate those concepts.
> In addition, our work has implications for lawsuits and copyright compensation for artists whose training data was used to train models. The damage can be estimated by auditing companies that train such models to determine whether the frequency of their training images falls above or below the threshold.
>
> > 2. what is the primary driver of these threshold differences between SD2.1 and SD1.5 — dataset size or text encoder architecture?
>
> Our results indicate that the difference in the thresholds between SD2.1 and SD1.5 is due to the text encoders. This is because even though all models in SD series 1 are trained on different datasets (https://huggingface.co/stable-diffusion-v1-5/stable-diffusion-v1-5), they have the same imitation threshold ((Line 353 in Section 6), while SD2.1 has a different threshold. Notably all models in SD series 1 have the same text encoder, while the text encoder of SD2.1 differs. We provide the full experimental details in Appendix G of the paper and our conclusion has also been shared  by other practitioners: https://www.assemblyai.com/blog/stable-diffusion-1-vs-2-what-you-need-to-know/
>
> > Moreover, the paper does not explore other potentially influential factors, such as model size, which limits the comprehensiveness of the analysis.
>
> Our approach requires access to open-source models whose training data is available (we conducted experiments with all such models in the paper). The models that satisfy this criterion vary in factors like model sizes, training data, and text encoders. Therefore it is not possible to ablate the effect of each factor separately. The paper’s methodology is applicable to any model/data pairs in the future that can help to separate out the effects of all factors affecting the threshold once such models become available.
>
> To test whether model size could affect the imitation threshold, we conducted experiments with the smaller Latent Diffusion model trained on LAION-400M dataset and found the thresholds to be similar to the ones found for SD1.1, SD1.5, SD2.1. We have added this experiment in Appendix A in the updated paper.

---

> > ### Comment · Reviewer_97Vw · 2024-12-03
> >
> > Thank you for your response and the additional details and experiments included in Appendix A. Based on this effort, I have slightly increased my score.
> >
> > I appreciate your explanation regarding the proposed imitation threshold, which indeed offers meaningful insights that could advance both conceptual understanding (e.g., detecting infringement) and practical applications (e.g., developing unlearning approaches). However, the primary contribution of the paper -- the identification of a threshold range of 200–600 images across two domains -- lacks detailed elaboration and evidence on how this threshold can be concretely applied.
> >
> > The impact of this work would be significantly enhanced by elaborating on real-world cases and practical considerations, as seen in your reference to related legal cases in response to Reviewer hEno. Additionally, including preliminary results demonstrating how computing the proposed threshold benefits representative imitation mitigation models (e.g., those outlined in Appendix B) would make the contribution more solid and actionable.

---

> > > ### Author Response · Authors · 2024-12-04
> > > **Author Response**
> > >
> > > We thank the reviewer for reading our rebuttal and increasing your score in light of it.
> > >
> > > > the identification of a threshold range of 200–600 images across two domains -- lacks detailed elaboration and evidence on how this threshold can be concretely applied. The impact of this work would be significantly enhanced by elaborating on real-world cases and practical considerations, as seen in your reference to related legal cases. Additionally, including preliminary results demonstrating how computing the proposed threshold benefits representative imitation mitigation models (e.g., those outlined in Appendix B) would make the contribution more solid and actionable.
> > >
> > >
> > > Our work aimed to find the imitation threshold that can be useful for both a) developers of text-to-image models who want to avoid copyright and privacy infringement (L. 52, 539) (by lowering the image count of a concept to keep it below the imitation threshold) and b) victims of copyright infringements, who can measure the imitation between the generated artworks and their real artworks and use our method to estimate the number of the copyrighted images in the training data of the model -- which can help them get compensation (L50., 539).

---

### Official Review · Reviewer_WqCq · 2024-11-04

**Soundness:** 2
**Presentation:** 2
**Contribution:** 2
**Rating:** 5
**Confidence:** 4

**Summary:**

The paper makes the following claims:
- Relationship between a "concept's" frequency in training dataset and the ability of a text to image model to imitate it. They find the threshold to be 200-600 images
- Propose an efficient approach to estimate the imitation threshold



I have the following questions and suggestions. I am willing to change my review based on the authors' response.

1. The paper does not define "concepts," assuming they are either self-explanatory or domain-specific. This assumption introduces potential bias or error in the paper's claims.
    - How distinct should concepts be from one another to qualify as separate concepts? How is the distance between concepts computed? For instance, many artists share "styles" due to shared lineage, schools, or subjects, which can significantly impact the imitation threshold. Broad art styles like "Avant-Garde," "Deco," or "Baroque" are loosely defined. Depending on how a concept is defined, the result of 200-600 images could vary significantly.

    - Many concepts, such as art styles, overlap and are composite. For example, Cubism, Futurism, and Constructivism share many features, as do Realism, Photorealism, and Precisionism, or Expressionism, Fauvism, and Die Brücke. Furthermore, concepts like "cubist impressionism" merge cubism and impressionism. The paper does not provide details on defining a concept or its boundaries, making it difficult to justify the main motivation and claim of efficiently estimating the imitation threshold of a "concept." In cases of composite and overlapping concepts, learning one concept may transfer to another, leading to erroneous imitation thresholds. This makes it hard to justify the paper's main motivation and claim (L91-95, L21-25) of that of efficiently estimating the imitation threshold of a "concept".


1. L148-151 - How do the authors make sure that the increasing number of samples of a particular concept is *causing* the models to learn to imitate? For example, how do we discount the case that if we train a diffusion model on 1 million concepts with each concept represented by 100 images (well below the imitation threshold of any individual concept), the model may still be able to imitate a particular concept due to an overall increase in general image generation understanding? Again, this relates to the main claim of the paper (L91-95, L21-25) of that of efficiently estimating the imitation threshold of a "concept." The paper seems to address a causal question without providing sufficient causal analysis.

1. The assumptions of the paper are too strict, which makes the method to be of much practical relevance.
    - Particularly, the following two assumptions are idealistic: distributional invariance between the images of all concepts in a domain and no confounders between the imitation score and image count. Can the authors justify these assumptions? Further, can the authors show experiments upon loosening these assumptions?
    - Can we introduce error bounds to account for the strict assumptions? This is important, especially since the abstract and introduction both try to answer a legal question (L24-25, L36-39) with a very prescriptive answer in the abstract, introduction, and results section (200-600 images in general and specific numbers like 364 and 234 images for different models in L342-355) without contextually mentioning the strict assumptions under which their analysis holds true or mentioning the error bounds. One gets to know the limiting assumptions, only when one reads the paper in details. While this may not be author's intention, but this gives a false assumption about the correctness/error bounds of the various imitation threshold numbers.


1. Another (strict) assumption not listed in the assumption section is L240-242 "prompts are distinct from the captions used in the pretraining dataset to minimize reproduction of training images". (Since the paper mentions the case of StabilityAI) Interestingly, In the famous New York Times vs. OpenAI lawsuit, the lawyers tested GPT, especially flouting this assumption. They start the prompt with the first few words of New York Times articles and let GPT complete the rest of the article [1]. Their case mentions many examples where exact reproduction is observed. This is an example of an imitation. How does the paper address such (actual) cases of imitations when discussing the "imitation threshold"?  Further, this assumption should be noted in the assumption section and when claiming prescriptive thresholds of 200-600 images.



1. In Sec-5.2, I could not understand why the same classifier couldn't be used as was trained in Sec-5.1 (potentially, along with the other classifiers the paper uses).




[1] https://nytco-assets.nytimes.com/2023/12/NYT_Complaint_Dec2023.pdf

**Strengths:**

Mentioned in the summary section

**Weaknesses:**

Mentioned in the summary section

**Questions:**

Mentioned in the summary section

---

> ### Author Response · Authors · 2024-11-20
> **Author Response**
>
> We thank the reviewer for taking the time and effort to review our paper and we are glad that the reviewer found our proposed approach efficient to estimate the imitation threshold. We address the limitations as follows:
>
> > 1. The paper does not define "concepts," assuming they are either self-explanatory or domain-specific. How distinct should concepts be from one another to qualify as separate concepts? How is the distance between concepts computed? The paper does not provide details on defining a concept or its boundaries
>
> We define a concept as a specific entity like a person or a specific person’s art style (L46 in Section 1). For the case of faces, each person’s face is considered a distinct concept. For the case of art styles, each artist’s art style is considered a distinct concept and we measure imitation specifically with respect to the artworks that a particular artist has produced (instead of measuring imitation with respect to a group of art styles like Cubism or Futurism). We have added this clarification in the paper (L. 213-214).
>
> The only requirement for an entity to be a valid concept is if the embedding models are able to distinguish it from other concepts (which is important for accurately measuring a concept’s frequency and imitation score). We use domain-specific embedding models (one for faces and another for art style) that are able to distinguish between concepts accurately. We conduct experiments to empirically verify this and present the results in Figure 12 (a) and (b) in Appendix F: we observe that the embedding similarity between artworks of the different artists is much lower than the similarity between the artworks of the same artist -- indicating that our embedding model can separate the art styles of different artists accurately, and these art styles are a valid set of concepts. In Figure 7 in Appendix E, we present the results for the embedding similarity between the faces of the same and different individuals. We find that the similarity between faces of different individuals is much lower than the similarity between faces of the same individual -- indicating that the embedding model distinguishes between faces of different individuals accurately.
>
> We hope that these experimental results allay the reviewer’s concern about how a valid concept is selected. We note that using a different set of embeddings models can change the set of concepts one can use for the experiment.
>
> > In cases of composite and overlapping concepts, learning one concept may transfer to another, leading to erroneous imitation thresholds.
>
> As our experimental results show in Figure 7 and Figure 12, the embedding models can accurately distinguish between the different concepts we experiment with, the individual faces and art styles. Therefore even if there is an overlap between certain concepts (e.g. multiple art styles of Post-impressionism kind), we can accurately measure the frequency and imitation score for each concept, alleviating concerns about the overlap between different concepts.
>
> > 2. How do the authors make sure that the increasing number of samples of a particular concept is causing the models to learn to imitate?
>
> To establish causal relationship there are two major approaches: RCTs and using observational data to estimate the effect. While RCTs in our case would be extremely expensive (L 150-151), we are able to estimate such effects using the observational data approach.
> As such, we assume there are no confounding factors that affect both the frequency of a concept and its imitation score. Under this assumption, from our experimental plots, the frequency of a concept causally affects its imitation score (an increase in frequency increases the imitation.) This intuition has also been verified by previous works which argue about a concept’s frequency and a model’s ability to perform tasks related to those concepts (such as image classification, image generation, or text retrieval). All these works find that as the number of samples of a particular concept increases, it increases the ability of the model to perform these tasks. No "Zero-Shot" Without Exponential Data: Pretraining Concept Frequency Determines Multimodal Model Performance [https://arxiv.org/abs/2404.04125] show this relation for image classification and generation tasks, and Large Language Models Struggle to Learn Long-Tail Knowledge [https://arxiv.org/abs/2211.08411] show this for question answering tasks. Large Language Models Struggle to Learn Long-Tail Knowledge even performs causal experiments to establish that frequency indeed affects model’s performance for that concept. Therefore based on these past works, we can claim that increasing the number of samples of a concept would increase the ability of a text-to-image model to imitate it.

---

> ### Author Response · Authors · 2024-11-20
> **Authors Response Continued**
>
> > For example, how do we discount the case that if we train a diffusion model on 1 million concepts with each concept represented by 100 images the model may still be able to imitate a particular concept due to an overall increase in general image generation understanding?
>
> We would like to address this misunderstanding. We designed our setup to find the imitation threshold for text-to-image models that are trained with real world datasets. In this setup, we find that concepts lower than the threshold are unlikely to be imitated. Our setup cannot be used to deduce the imitation threshold for this scenario of each concept having the same number of images, as it is significantly different from our setup. Therefore, we do not claim our imitation thresholds would transfer to all model/data pairs (L 520-522 in Section 8).
>
> > 3. The assumptions of the paper are too strict, which makes the method to be of much practical relevance. Can the authors justify these assumptions? Further, can the authors show experiments upon loosening these assumptions?
>
> The assumptions we make are standard when answering causal questions using observational data and have been made in several previous works: Pearl (2009); Lesci et al. (2024); Lyu et al. (2024). We have also empirically verified the validity of the assumptions, specifically the distribution invariance assumption between the concepts. If this holds true, then the imitation scores of two concepts (from the same domain) whose image counts are similar, should also be similar. To test whether this is empirically true for the domains we experiment with, we measure the difference in the imitation scores for concepts whose image counts differ by less than 10 images and report the difference averaged over all such pairs. The table below shows that the average difference in the imitation scores for all the pairs, for all the datasets we experiment with, is very close to 0. This provides empirical support for the validity of our assumptions. We also present this in Appendix B (Appendix C in the updated paper).
>
> | Domain | Dataset  | Avg. difference in imitation score |
> |--|----|---|
> | Human Faces| Celebrities | 0.0007 |
> | Human Faces| Politicians | 0.0023 |
> | Art Style | Classical Art Style | -0.0088 |
> | Art Style | Modern Art Style | -0.0013 |
>
> > Can we introduce error bounds to account for the strict assumptions?
>
> Thank you for this suggestion! In order to introduce error bounds on the imitation thresholds, we conduct an experiment in which we randomly sample 300 concepts from the set of 400 concepts for each domain we experimented with, and estimate the imitation threshold for each sampled set. We repeat this sampling process 1000 times for each domain and present the mean and variance of the imitation thresholds for each model. This experiment simulates a scenario where the invariance assumption holds only for a subset of the 400 concepts.
>
> | Pretraining Dataset | Model | Celebrities | Politicians| Classical | Modern|
> |----|-------|--|---|---|-----|
> | LAION2B-en | SD1.1 | 399 $\pm$ 85 | 284 $\pm$ 87 | 304 $\pm$ 56 | 208 $\pm$ 26 |
> | LAION2B-en | SD1.5 | 371 $\pm$ 23 | 302 $\pm$ 113 | 302 $\pm$ 65 | 212 $\pm$ 29 |
> | LAION-5B | SD2.1| 617 $\pm$ 148 | 385 $\pm$ 117 | 330 $\pm$ 142 | 292 $\pm$ 107|
>
> We find that for almost all domains and models the imitation thresholds we report in the paper are within the one standard deviation of the mean threshold this experiment finds. Our original claim about 200-600 images being required for concept imitation holds with this experiment as well. We updated the paper with this new experiment in Appendix C (L. 921-937)
>
> > 4. Another assumption not listed in the assumption section is "prompts are distinct from the captions used in the pretraining dataset to minimize reproduction of training images". Interestingly, the famous New York Times vs. OpenAI lawsuit mentions many examples where exact reproduction is observed.
>
> We would like to highlight that this is not an assumption of our work, but a core property of our research question. In this work, we are not tackling the question of memorization (which was well studied in previous works, e.g., L 812-814), but instead, are interested in the question of imitation, which is under-studied.
> The NYT vs. OpenAI lawsuit would correspond to an exact replication of a training image by a text-to-image model (if the lawsuit was related to text-to-image models). Therefore using “prompts distinct from the captions in the pretraining dataset” is an intentional choice rather than an assumption of the paper (because we want to measure imitation and not memorization). We choose to use different prompts as this was suggested by Somepalli et al. [https://arxiv.org/abs/2305.20086] to prevent generation of memorized training images, and therefore if a concept is generated by a model when using the different prompts, we can be confident about the imitation of the concept and not memorization of a training image.

---

> > ### Author Response · Authors · 2024-11-20
> > **Author Response Continued**
> >
> > > 5. In Sec-5.2, I could not understand why the same classifier couldn't be used as was trained in Sec-5.1 (potentially, along with the other classifiers the paper uses).
> >
> > We use the same embedding models in Section 5.1 and Section 5.2 (although for different tasks - in Section 5.1, the embedding models are used to count the number of training images of a concept and in Section 5.2, the embeddings models are used to accurately measure imitation between generated and training images). We use InsightFace as the face embedding model (Deng et al., 2020) and we use CSD as the art style embedding model (Somepalli et al., 2024). L. 276-277 and L. 311-314 in Section 5 mention them.

---

> > > ### Comment · Reviewer_WqCq · 2024-11-22
> > >
> > > > we measure imitation specifically with respect to the artworks that a particular artist has produced
> > >
> > > Despite this clarification, my original concern still holds. An artist's style is not unique. It is, more often than not, a combination of multiple styles. Further, artists are often proponents of specific art styles. For instance, Picasso is famous for his Cubism style.
> > >
> > > Practically, artist style, as opposed to art styles, are even more problematic because of the exact numbers (e.g, 364 images and 234 images) presented in the paper without ranges or error bounds. My concern with this framing is that such unambiguous and absolute numbers will be cited by lawyers to claim that the diffusion models imitate a concept or not. See  (L21-25, L36-40, L47-51, L93-95, L105-113, L210-215 L429-446, L528-539) as an example where the paper discusses conclusions mentioning privacy, attribution, and copyright laws. Nowhere in the paper are there any error bounds or a discussion of the strict assumptions under which these numbers are valid or the fact that there are several sources of uncertainty in the framing (for instance, classifier accuracies). Despite this, the numbers are presented as highly confident measures and singular truths for measuring imitation thresholds. This is a misrepresentation of experimental conditions under which these numbers are calculated.
> > >
> > > I appreciate that the authors have introduced a new error analysis in their rebuttal. Thank you for this effort. However, I found the explanation unclear. Could you elaborate on the exact steps taken in your experiment? Additionally, please revise the abstract, introduction, and discussion sections to reflect these error bounds and assumptions more explicitly. This would significantly strengthen the paper by providing a more nuanced and accurate representation of its claims.
> > >
> > > I also strongly recommend incorporating an uncertainty analysis and including error bounds throughout the work. This would help moderate the claims made in the abstract, introduction, and discussion sections, ensuring they align with the limitations and assumptions of the methodology.
> > >
> > > Finally, I note that the paper frequently references terms such as "law," "privacy," "copyright," "lawsuits," "infringement," and "attribution" (e.g., L21-25, L36-40, L47-51, L93-95, L105-113, L210-215, L429-446, L528-539, among others). However, it does not adequately discuss the assumptions and uncertainties under which its findings can or cannot be applied in legal contexts. I strongly suggest adding a discussion section clarifying when the results can legitimately inform legal cases and when they cannot.
> > >
> > >
> > > Regarding the statement: "In this work, we are not tackling the question of memorization"—this point is crucial and should be explicitly mentioned in the introduction. If the paper references the Stable Diffusion lawsuit to support its arguments, it is equally important to address the OpenAI lawsuit and clearly state that this work does not resolve the broader question of memorization.
> > >
> > > Given the paper's apparent goal of addressing legal questions—indicated by the frequent references to law-related terms (over 70 mentions)—it is vital to address the technical details and limitations that impact the reliability of its legal conclusions.
> > >
> > >
> > > > We use the same embedding models in Section 5.1 and Section 5.
> > >
> > > Please clarify this in the paper.

---

> > > > ### Author Response · Authors · 2024-11-23
> > > > **Author Response to the Comments**
> > > >
> > > > Thank you for your prompt response. We value your feedback and have incorporated your suggestions in the updated draft. Please see our responses and details to your suggestions below.
> > > >
> > > > >  An artist's style is not unique. It is, more often than not, a combination of multiple styles. Further, artists are often proponents of specific art styles. For instance, Picasso is famous for his Cubism style.
> > > >
> > > > While we agree with the reviewer that an artist’s style can be a combination of multiple styles and often belong to art style families like Cubism, the goal of our approach was to use a classifier that can distinguish between art styles of different artists. And our classifier is indeed able to do that. We show this by measuring the similarity between the embedding of the art works of 400 artists (when the images of these artworks are embedded by the art style model, CSD, we use). Figure 12 (a) and (b) in the Appendix shows the histograms of similarities between these artworks of different artists. We show that artworks by different artists are highly separable even when they belong to the same art style family (like Cubism or Impressionism).
> > > > The flexibility of our framework and research question would allow future work to ask the questions the reviewer is interested in, given an appropriate embedding model that is able to separate between different art style families.
> > > >
> > > > > I appreciate that the authors have introduced a new error analysis in their rebuttal. Thank you for this effort. However, I found the explanation unclear. Could you elaborate on the exact steps taken in your experiment?
> > > >
> > > > We conduct experiments to simulate a situation when the distributional invariance assumption between the concepts in a domain is not met. Specifically, we consider a situation where the assumption does not hold for all 400 concepts in a domain, but only for a subset of them and we only use this subset to find the imitation threshold.
> > > >
> > > > To estimate the error bounds we perform a permutation test where we sample `n` concepts for each domain and dataset at a time. We then find the imitation threshold using these `n` concepts and repeat this process 1,000 times by randomly sampling `n=300` concepts, which yields a distribution over the imitation thresholds per domain/dataset. We report the mean and standard deviation over the imitation thresholds in Table 3. We have added this to Section 5.3 in the updated draft of the paper.
> > > >
> > > > > … I also strongly recommend incorporating an uncertainty analysis and including error bounds throughout the work.
> > > >
> > > > We have changed the imitation thresholds in Table 3 to be the mean and one standard deviation of the imitation thresholds we compute using the error analysis experiment we conducted (described below). We added a mention to the error bounds on the thresholds, as well a more hedged discussion of the imitation threshold in the abstract, introduction, formulation, results, and the discussion sections in the updated paper. The changes are highlighted in the updated draft of the paper.
> > > >
> > > > > I strongly suggest adding a discussion section clarifying when the results can legitimately inform legal cases and when they cannot.
> > > >
> > > > We added a discussion about how to interpret our results in the updated paper, explicitly addressing the considerations on legal actions based on our results in Section 8 on L. 520-524.
> > > >
> > > > > Regarding the statement: "In this work, we are not tackling the question of memorization"—this point is crucial and should be explicitly mentioned in the introduction.
> > > >
> > > > We clarified this point in L. 43-46 in the introduction section of the updated paper.
> > > >
> > > > > We use the same embedding models in Section 5.1 and Section 5. Please clarify this in the paper.
> > > >
> > > > We clarified this point in L. 304-305 in Section 5.2 in the updated paper.
> > > >
> > > >
> > > >
> > > >
> > > > _Thank you again for your valuable feedback, which has improved the readability and statistical strength of our paper! We hope our answers have satisfied your concerns, and we would love to continue this conversation otherwise._

---

> > > > > ### Comment · Reviewer_WqCq · 2024-11-24
> > > > >
> > > > > 1. L015-016 "seek to determine the point at which a model was trained on enough instances to imitate a concept – the imitation threshold." - Current error bounds indicated in the author's reply are between 20%-45%. From this lens, I do not think that the imitation threshold is a point. It is a range.
> > > > >
> > > > > 1. To validate the claim "Therefore, if the imitation threshold is not reached, infringement is unlikely"
> > > > > I think it would be valuable to show an experiment where a model is trained with images equal to the imitation threshold and 1 below and above the threshold's error bounds. This should be shown for at least a few concepts. The current Fig.2 is inadequate for this since it shows this over a very wide variance - from 3 to 100 times 3 to 1000 times 3 and so on. Please keep the concept consistent in the experiment.
> > > > >
> > > > > 1. I feel that my initial comment "Further, can the authors show experiments upon loosening these assumptions?" is still unanswered. There are four core assumption:
> > > > >   - No memorization, only imitation
> > > > >   - Distributional invariance
> > > > >   - no confounders between the imitation score and the image count of a concept
> > > > >   - each image of a concept contributes equally to the learning of the concept
> > > > >
> > > > > I think it would be very valuable to see (even small-scale) experiments on loosening each of these (strict) assumptions and seeing the impact of this on imitation thresholds. If there are computational constraints to run it over bigger models, I would welcome some experiments on smaller models with fewer concepts. This particular concern is important for me to address because it pertains to the main contribution of the paper ("we formalize a new problem – Finding Imitation Threshold", "we propose a method to efficiently estimates the imitation threshold for text-to-image models", "We seek to determine the point at which a model was trained on enough instances to imitate a concept", "The imitation threshold can provide an empirical basis for copyright violation claims and acts as a guiding principle for text-to-image model developers that aim to comply with copyright and privacy laws".)
> > > > >
> > > > > Also, the updated paper says "including tight error bounds for each setting". Can you please formally define the tightness constraints in the paper?
> > > > >
> > > > >
> > > > >
> > > > > 4. Please make the naming consistent - "art style" or "artist's style". In the author's reply, they said "we measure imitation specifically with respect to the artworks that a particular artist has produced" where as L20 says "art styles".
> > > > >
> > > > > > Given the paper's apparent goal of addressing legal questions—indicated by the frequent references to law-related terms (over 70 mentions)—it is vital to address the technical details and limitations that impact the reliability of its legal conclusions.
> > > > >
> > > > > 5. Finally, please add a section in the appendix addressing this concern. Please add the core assumptions of this work in non-technical language, a brief note of how the imitation thresholds are calculated, and when can a lawyer or model developer rely on these thresholds and when can they not.

---

> > > > > > ### Author Response · Authors · 2024-12-03
> > > > > > **Author Response**
> > > > > >
> > > > > > Thank you again for your prompt response. We value your feedback and have incorporated your suggestions in the updated draft and also conducted the experiments you suggested. Please see our responses and details to your suggestions below.
> > > > > >
> > > > > > > I think it would be valuable to show an experiment where a model is trained with images equal to the imitation threshold and 1 below and above the threshold's error bounds. This should be shown for at least a few concepts. Please keep the concept consistent in the experiment.
> > > > > >
> > > > > > Thank you for suggesting this. We conduct this experiment for face imitation for 10 celebrities. We do this by finetuning a pre-trained SD model (SD1.4) on an increasing number of images of each of these 10 celebrities, starting with 50 images (much below the threshold) and going up to 800 images (above the threshold). To match the training procedure of this model we construct a training dataset with 10K random LAION images including a specific number of images for a specific concept. This finetuning experiment is the closest we can come to training a model on a concept that the model has not seen before and computing the imitation threshold in an optimal manner (“optimal approach” as mentioned in Section 3). This method of finding the imitation threshold also **does not require** the two assumptions we make: a) distributional invariance of the images in the concepts, and b) absence of confounders between the imitation score and the image count. This is because a) the finetuning is done on each concept individually and b) the celebrities were selected randomly.
> > > > > >
> > > > > > The table below shows the average imitation score for these 10 celebrities when a model is finetuned on a different number of their images. Note that the score for imitation to occur for this model is 0.26 (this was the average imitation score on the right side of the threshold in Figure 16 in the Appendix). And from the table below we notice that this imitation score is reached when the number of finetuned images is 200-400. The score is lower than the required score for imitation (lower than 0.26) when the model is finetuned with 50 or 100 images (which are below the threshold) of a celebrity. Note that the number of images required in this experiment is very close to the imitation threshold MIMETIC^2 estimates for this model (234 as shown in Figure 16 in the Appendix). We will add this finetuning experiment for other models we experiment with for the camera-ready paper.
> > > > > >
> > > > > > | Images | Imitation Score After Finetuning | Imitation Occurs |
> > > > > > |:------:|:-------:|:-------:|
> > > > > > | 0      | 0.03    | No |
> > > > > > | 50     | 0.14    | No |
> > > > > > | 100    | 0.18    | No |
> > > > > > | 200    | **0.27**    | Yes |
> > > > > > | 400    | **0.34**    | Yes |
> > > > > > | 600    | **0.39**    | Yes |
> > > > > > | 800    | **0.42**    | Yes |
> > > > > >
> > > > > > > I think it would be very valuable to see (even small-scale) experiments on loosening each of these (strict) assumptions and seeing the impact of this on imitation thresholds.
> > > > > >
> > > > > > The finetuning experiment we show above finds the imitation thresholds without requiring two key assumptions we make in the paper: a) distributional invariance of the images in the concepts, and b) absence of confounders between the imitation score and the image count. However, it still does make the assumption regarding each image of a concept contributing equally to its learning. Since our research question is about the number of images of a concept, we do not aim to get rid of this assumption, similar to most sample complexity works (Valiant, 1984; Udandarao et al., 2024; Wen et al., 2024; Wang et al., 2017).
> > > > > >
> > > > > > > Also, the updated paper says "including tight error bounds for each setting". Can you please formally define the tightness constraints in the paper?
> > > > > >
> > > > > > What we meant by the tight error bounds were the error bounds we report in Table 3 in the paper, which are computed empirically and were found to be narrow. We did not  mean “tight” to refer to the error bounds with respect to the ground truth imitation thresholds (we do not have access to them). To improve clarity, we have removed the word “tight” in the updated paper draft (L. 94).
> > > > > >
> > > > > > > Please make the naming consistent - "art style" or "artist's style".
> > > > > >
> > > > > > We have revised all mentions of “artist’s style” to “art style” to be consistent in the updated draft of the paper.
> > > > > >
> > > > > > > From this lens, I do not think that the imitation threshold is a point. It is a range.
> > > > > >
> > > > > > The imitation threshold we estimate is a specific value. To provide further confidence in such a value, we also estimate an error bar that indicates the confidence of that value to be within the range of one standard deviation. The updated Table 3 in the paper also reflects this with the imitation threshold being a mean value with one standard deviation.

---

> > > > > > > ### Author Response · Authors · 2024-12-03
> > > > > > > **Author Response Continued**
> > > > > > >
> > > > > > > > please add a section in the appendix addressing this concern. Please add the core assumptions of this work in non-technical language, a brief note of how the imitation thresholds are calculated, and when can a lawyer or model developer rely on these thresholds and when can they not.
> > > > > > >
> > > > > > > Thank you for suggesting that. We have added another section in the Appendix (Appendix C, L. 850-875) to explain the assumptions and other technicalities in a non-technical language, which a lawyer or model developers can use in the updated paper draft. This is in addition to a paragraph that we added in the discussion section about this (L520-524), based on your earlier suggestion, and we point to this new Appendix C in this part of the paper.
> > > > > > >
> > > > > > > Since we cannot upload a new paper draft, we copy the content of Appendix C here:
> > > > > > >
> > > > > > > Our work estimates the imitation thresholds under certain assumptions: distributional invariance between images of different concepts and absence of confounders between the imitation score and image count of a concept. We believe these assumptions are reasonable (as we argue in Section 3 and empirically demonstrate in Appendix D) and they have also been made in several prior works. However, they might limit the direct transfer of the imitation thresholds we report to other settings, and more research should be made to validate our results to different settings.
> > > > > > >
> > > > > > >
> > > > > > >
> > > > > > >
> > > > > > > _Thank you once again for your valuable comments and suggestions, and engaging with us during this discussion period. This continued discussion has strengthened our paper substantially. We hope our answers have satisfied your concerns, and we hope you can update your score accordingly._

---

> > > > > > > > ### Author Response · Authors · 2024-12-04
> > > > > > > > **Author Response Continued**
> > > > > > > >
> > > > > > > > In lieu of the reviewer's earlier request to conduct experiments with any of the current assumptions -- for which we already showed the finetuning experiment that got rid of the first two assumption, we performed another experiment to also test the validity of the third assumption: equal image contribution assumption.
> > > > > > > >
> > > > > > > > For this experiment, we finetune a SD model on images of concepts randomly sampled from a large pool of images and measure the imitation score of the finetuned model on this concept. We report the variance in the imitation scores across the random sampling of the training images. If the variance is low we can conclude that our “equal image effect” assumption (Section 3 and 8) is valid. We performed this experiment for 5 concepts and finetuned SD 1.4 model on 400 images of each of these concepts. The table below shows the mean imitation score and the standard deviation in the scores. We can see that the std. deviations are very small for all concepts, validating our assumption.
> > > > > > > >
> > > > > > > > We will add extended experiments to the camera-ready version, but these results suggest this assumption is valid as well.
> > > > > > > >
> > > > > > > > | Celeb                     | Mean Imitation Score | Std. Deviation |
> > > > > > > > |:-------------------------:|:--------------------:|:--------------:|
> > > > > > > > | Javier Milei              | 0.36                 | 0.02           |
> > > > > > > > | Hashim Safi al-Din        | 0.38                 | 0.03           |
> > > > > > > > | Shyam Rangeela            | 0.31                 | 0.03           |
> > > > > > > > | Akshata Murty             | 0.30                 | 0.03           |
> > > > > > > > | Hana-Rawhiti Maipi-Clarke | 0.30                 | 0.02           |

---

### Meta-Review · Area_Chair_jXb6 · 2024-12-20

**Metareview:**

**Summary**

This work investigates the relationship between the frequency of a concept in the training data of text-to-image models and the model’s tendency to imitate that concept. The authors propose an approach, MIMETIC^2, to estimate the imitation threshold, which is the approximate number of instances of a concept needed for the model to have a tendency to imitate it. Through empirical analyses, they provide estimates of this threshold for three different Stable Diffusion models.

**Strengths**

- The reviewers note that the authors address an under-explored but important phenomenon: How many training examples are needed before a model begins to imitate a concept reliably.
- The paper is logically structured, clearly outlining the problem, assumptions, and analyses.
- The authors acknowledge limitations and suggest paths for future research, noting the complexity of defining concepts and establishing meaningful thresholds.

**Weaknesses**

- The primary concern is the lack of a clear, operational definition of “concept.” Since many concepts (styles, genres) can overlap or be ill-defined, the reported thresholds may not be reliable or generalizable.
- The restrictive assumptions used in the analysis lack sensitivity tests. Without exploring how results change when these assumptions are relaxed, the applicability and reliability of the thresholds are questionable.
- The reviewers find the practical impact and interpretability of the reported thresholds unclear. It remains uncertain how these findings could guide mitigation strategies or clarify which factors truly influence the imitation threshold.
- The scope of the experiments is limited. The reviewers suggest examining a broader range of models and datasets, as well as addressing outliers or overlapping concepts.

**Decision**

After considering the authors’ revisions, some reviewers acknowledged improvements and slightly raised their scores. However, the key issues remain unresolved, including the unclear definition of concepts, insufficient demonstration of practical applicability, and limited experimental scope. The reviewers’ final recommendations converge on a marginal evaluation (5: Marginally below threshold, 5: Marginally below threshold, 6: Marginally above threshold). Based on these assessments, I recommend rejection of this paper.

**Additional Comments On Reviewer Discussion:**

Despite the authors’ attempts to clarify their methods and provide additional details in the rebuttal, reviewers remained concerned about the paper’s core issues. The reviewers noted that the concept definition lacked specificity, the sensitivity of the reported thresholds was insufficiently tested, and the practical relevance of the findings was unclear. While the authors offered some additional experiments and sensitivity analyses, these were seen as inadequate to resolve the fundamental limitations. As a result, the reviewers’ evaluations, though slightly improved in some cases, still converged on a marginal recommendation, leading to a final decision to reject the paper.

---

### Decision · Program_Chairs · 2025-01-22

Reject